# Effect of Exogenous Melatonin Application on the Grain Yield and Antioxidant Capacity in Aromatic Rice under Combined Lead–Cadmium Stress

**DOI:** 10.3390/antiox11040776

**Published:** 2022-04-13

**Authors:** Ye Jiang, Suihua Huang, Lin Ma, Leilei Kong, Shenggang Pan, Xiangru Tang, Hua Tian, Meiyang Duan, Zhaowen Mo

**Affiliations:** 1State Key Laboratory for Conservation and Utilization of Subtropical Agro-Bioresources, College of Agriculture, South China Agricultural University, Guangzhou 510642, China; jiangyechena@163.com (Y.J.); 13668961353@163.com (S.H.); malin20220321@163.com (L.M.); panshenggang@scau.edu.cn (S.P.); tangxr@scau.edu.cn (X.T.); tianhua@scau.edu.cn (H.T.); scdmy213@163.com (M.D.); 2Scientific Observing and Experimental Station of Crop Cultivation in South China, Ministry of Agriculture and Rural Affairs, Guangzhou 510642, China; 3Guangzhou Key Laboratory for Science and Technology of Fragrant Rice, Guangzhou 510642, China; 4Rice Research Institute, Guangdong Academy of Agricultural Sciences, Guangzhou 510640, China; kongleilei1991@163.com

**Keywords:** antioxidants, cadmium–lead, aromatic rice, melatonin, microbial community structures

## Abstract

This study aimed to determine the mechanism of exogenous melatonin application in alleviating the combined Pb and Cd (Pb-Cd) toxicity on aromatic rice (*Oryza sativa* L.). In this study, a pot experiment was conducted; two aromatic rice varieties, Yuxiangyouzhan and Xiangyaxiangzhan, were selected, and sprays using 50, 100, 200, and 400 μmol L^−1^ melatonin (denoted as S50, S100, S200, and S400) and irrigation using 100, 300, and 500 μmol L^−1^ melatonin (denoted as R100, R300, and R500) were also selected. The results showed that, under the S50, S100, and S200 treatments, the Pb content of aromatic rice grain decreased, and the grain yield increased significantly. Moreover, the application of exogenous melatonin significantly reduced the accumulation of H_2_O_2_ in rice leaves at maturity under Cd–Pb stress and reduced the MDA content in Xiangyaxiangzhan leaves. In addition, the microbial community structure changed significantly under S50 and R300 treatments. Some pathways, such as the synthesis of various amino acids and alanine, aspartate, and glutamate metabolism, were regulated by S50 treatment. Overall, melatonin application improved aromatic rice grain yield while reducing heavy metal accumulation by regulating the antioxidant capacity and metabolites in aromatic rice plants and altering the physicochemical properties and microbial community structures of the soil.

## 1. Introduction

Rice is a staple food worldwide, and aromatic rice is most popular due to its unique flavor [1]. With the improvement of living quality, people are paying more attention to the nutrition provided by food while ensuring a good taste. Although aromatic rice attracts more attention in trade marketing, the rice plant exhibits higher heavy metal bioaccumulation than other cereals, affecting its safety [2]. With industry development, heavy metal ions are inevitably released into farmland [3]. Plant tissues accumulate heavy metals once their concentration exceeds the potential limit in the soil [4]. For cereal crops, heavy metals could accumulate in the grain and eventually enter the human body along the food chain, threatening human health [5,6]. Therefore, strategies to mitigate the adverse effects resulting from heavy metals in rice crops, particularly in aromatic rice, are necessary to ensure food safety.

Cadmium (Cd) and lead (Pb), among other heavy metals, are the most serious, and agricultural soils commonly see combined Cd and Pb (Cd–Pb) pollution [7,8]. The inhibition effect of two or more heavy metals on plant growth is often more potent than that of single heavy metals [9]. Previous studies have indicated that the toxicity of both Cd and Pb leads to the excessive production of reactive oxygen species (ROS), which could disrupt cell cytomembrane and cause programmed cell death, directly or indirectly affecting various physiological processes in rice [10,11]. Under Cd–Pb stress, the growth of rice plants would be inhibited, leading to leaf curling and ultimately resulting in yield reduction and quality decline [12,13]. Roots are an essential part of plants to communicate with soil; thus, the effect of heavy metals on soil microorganisms has attracted more and more attention. A previous study reported that the incentive response of soil microorganisms to Cd–Pb contamination can help identify and reduce the ecological risk of heavy-metal-contaminated soils [14]. Most importantly, once Cd and Pb enter the rice, they are enriched in the rice grain, entering the food chain, eventually harming human health [5,15]. Thus, approaches in the soil–plant system are needed to establish simultaneous improvement in grain yield, grain quality, and grain safety.

Melatonin, chemically known as *N*-acetyl-5-methoxytryptamine, is an indole tryptamine [16]. Melatonin was first discovered in animals and has been proven to be involved in the physiological regulation of pathways in animals’ bodies [17]. In plants, melatonin, acting as the most vital antioxidant, is directly involved in the regulation of various physiological processes or otherwise serves as a signaling molecule to participate in the plant’s physiological regulation [18]. Reports suggest that melatonin enhances the antioxidant system in response to heat, salinity, and heavy metal stresses [19,20,21]. Melatonin has been reported to be a plant-growth regulator in relieving heavy metal stress in crops such as rice, maize, and wheat [22,23]. The growth, grain yield, and grain quality of crops can be modulated by melatonin application [19,24]. The endogenous melatonin level varies significantly between plants, with the highest level found in rice [17]. Exogenous melatonin application could regulate aromatic rice seedlings’ early growth and antioxidant enzyme activities in response to Cd toxicity [25], indicating that exogenous melatonin application is an effective approach to regulating grain yield and quality in rice.

Aromatic rice has a key aromatic compound; 2-acetyl-1-pyrroline (2AP) has undergone extensive identification in production [26]. Compared to ordinary rice, aromatic rice exhibits a lower grain yield and a more tolerant ability under adverse environmental conditions [27]. Previous studies have reported the effect of plant growth regulators, e.g., gamma-aminobutyric acid (GABA), on the regulation of grain yield and the fragrance of aromatic rice [28,29,30]. Studies have also reported the effect of Cd–Pb stress on aromatic rice production [4,31]. However, most studies focused on the regulatory effect of melatonin on other plants under single stress [32,33]. More importantly, most studies on the effects of exogenous melatonin on rice under stress have been conducted by the way of seed pretreatment, and it is not clear about the effects of the application exogenous melatonin to different parts of aromatic rice under Cd–Pb stress. Therefore, we used foliar spray and root irrigation for exogenous melatonin application. The present study aimed to (1) determine the Cd and Pb content of different rice tissues, as well as the grain yield of two aromatic rice cultivars under Cd–Pb stress; (2) determine the activity of antioxidant enzymes, including superoxide dismutase (SOD), peroxidase (POD), and catalase (CAT), and the content of glutathione (GSH), ascorbic acid (ASA), and metallothioneins (MTs); (3) evaluate the rhizosphere microbial diversity and measure leaf metabolites to access the function of exogenous melatonin on aromatic rice under Cd–Pb contamination; and (4) investigate the suitable exogenous melatonin application methods and concentrations for aromatic rice.

## 2. Materials and Methods

### 2.1. Experimental Design

A pot experiment was carried out on the teaching experimental farm of the South China Agricultural University (altitude: 11 m) from March to June in 2019. This region has a humid subtropical monsoon climate. The whole experiment was carried out under net house conditions. The average monthly temperature, the average monthly humidity, and the average monthly sunshine hours during the experiment are shown in Figure 1. Two rice cultivars, i.e., Yuxiangyouzhan and Xiangyaxiangzhan, which are well-known and widely grown aromatic rice cultivars in South China, were used in the study. The seeds of these cultivars were obtained from the Rice Research Institute of College of Agriculture, South China Agricultural University. Rice seeds were surface sterilized with 5% NaClO for 10 min and rinsed extensively in distilled water, then germinated for 48 h at 28 °C in darkness. Seeds were sown in a substrate by direct seeding and then transplanted at the two-leaf stage into a pot (diameter: 30 cm, height: 24 cm) containing 10 kg of dry paddy soil. Five holes were set in each pot, with five rice seedlings planted in each hole, which was checked often, to maintain the number of seedlings. The experimental soil was sandy loam, consisting of 15.06 g kg^−1^ organic matter, 0.83 g kg^−1^ total nitrogen (N), 1.10 g kg^−1^ total phosphorus (P), 20.10 g kg^−1^ total potassium (K), 2.08 mg kg^−1^ Cd, and 6.90 mg kg^−1^ Pb, with a pH value of 6.56. Ten grams of aromatic rice-specific fertilizer (12.5% N, 10% K_2_O, and 6% P_2_O_5_), used as the base fertilizer for both types of rice, was applied at 3 days before the transplanting in each pot. Moisture maintenance and pest and weed control were carried out according to routine management.

### 2.2. Experimental Treatments

This experiment used a randomized block design with five replicates (one-pot served as one replicate). Approximately 1000 kg of air-dried soil was taken, and CdCl_2_ and Pb(NO_3_)_2_ were added and mixed well to ensure that the mixture contained 2.08 mg kg^−1^ Cd and 6.90 mg kg^−1^ Pb to simulate a Cd–Pb stress environment. The application methods of exogenous melatonin included foliar spraying and root irrigation. Spraying treatments included various concentrations of melatonin: 50, 100, 200, and 400 μmol L^−1^ melatonin; the different melatonin concentrations were denoted as S50, S100, S200, and S400 treatments. Similarly, the different concentrations (100, 300, and 500 μmol L^−1^) of melatonin used for irrigation were denoted as R100, R300, and R500. In the rhizosphere microbial community structure and metabolite surveys, the treatment names of Xiangyaxiangzhan and Yuxiangyouzhan were preceded by X and Y. For each treatment, the first application was made at the rice heading stage (HS), and the set melatonin concentration was applied separately every 20 days for a total of 5 applications.

### 2.3. Sampling and Measurement

#### 2.3.1. Determination of Cd and Pb Contents

Five randomly selected rice seedlings were sampled at the HS and the MS. The root, stem leaf, and grain of the rice seedlings were separated and baked at 60 °C for 48 h to determine the Cd and Pb contents. The Cd and Pb contents in the plant tissues were determined according to Liu et al. [34]. Approximately 0.5 g of the dry sample was soaked in 4 mL of nitric acid and 3 mL of perchloric acid for 12 h and heated in a graphite furnace (HYP-340, Shanghai, China). The heating process consisted of four stages: 90 °C for 30 min, 140 °C for 40 min, 195 °C for 120 min, and 220 °C for 240 min. The filtered digesting solution was then placed in a 25-mL volumetric flask with a constant volume of 25 mL. Absorbent values were then determined by an atomic absorption spectrophotometer (GFA-EX7i), and the Cd and Pb contents were calculated according to the standard curve. The unit of Cd and Pb contents was expressed in mg kg^−1^. The translocation factor (TF) was calculated according to Huang et al. [13].

#### 2.3.2. Determination of Grain Yield and Yield Components

At the MS, three hills of rice plants were randomly selected from each pot, and four pots were taken as four replicates. After threshing and weighing, the actual yield of each pot was measured and calculated, and the productive tillers, grains per panicle, and 1000-grain weights were investigated.

#### 2.3.3. Determination of GSH, ASA, and MTs Contents

At the HS, 15 days after HS and the MS, fresh flag leaves were collected and stored at −80 °C till biochemical analyses. The GSH, ASA, and MTs contents in the plant tissues were determined according to Huang et al. [13]. Approximately 0.2 g of fresh plant tissue was homogenized in 4 mL of 5% TCA and centrifuged at 10,000× *g* for 15 min at 4 °C. The supernatant was used to determine the GSH, ASA, and MTs contents.

For GSH content determination, the 5 mL reaction system consisted of 0.2 mL of the supernatant, 4.4 mL of 0.1 mol L^−1^ phosphate buffer (PBS, pH 7.0), and 0.4 mL of the DTNB reagent (0.04%, *w*/*v*). The GSH content was expressed as μmol·g^−1^ fresh weight (FW).

For ASA content determination, the reaction system consisted of 0.4 mL of 150 mmol L−1 NaH_2_PO_4_ solution, 0.2 mL of enzyme extract, 0.2 mL of distilled water, and 0.4 mL of 10% TCA. After 30 s, the mixture was added to 0.4 mL of H_3_PO_4_, 0.4 mL of 4% 2, 2-bipyridine, and 0.2 mL of 3% FeCl_3_ solution. The amount of ASA in the sample was expressed as μmol g^−1^ FW.

For MTs content determination, 0.2 g of fresh plant sample was homogenized in 1 mL of grinding solution and centrifuged at 20,000 rpm for 60 min at 5 mL centrifuge tube for 70 °C 10 min water bath then centrifuged again at 20,000 rpm for 30 min at 4 °C. The grinding solution contained 0.02 mol L^−1^ Tris-HCl, 0.5 mmol L^−1^ PMSF, and 3 mL of 0.01% mercaptoethanol. The 5 mL reaction system consisted of 0.2 mL of extraction solution, 4.4 mL of demetalization reagent, and 0.4 mL of 0.04% DTNB solution, and PBS was taken instead of DTNB as a blank test tube. The reaction system was placed in the dark at 30 °C for 30 min to develop color. Then, the absorbance values were measured at 412 nm. The unit was expressed as μmol g^−1^ FW.

#### 2.3.4. Determination of SOD, POD, and CAT Activities and MDA and H_2_O_2_ Contents

At the HS, 15 days after the HS and MS, fresh flag leaves were collected and stored at −80 °C till biochemical analyses were performed. The antioxidant enzymes’ activity and the content of malondialdehyde (MDA) and H_2_O_2_ were measured following the previously reported methods [35]. The fresh leaf sample (0.3 g) was homogenized with 3 mL of 100 mM PBS solution and centrifuged at 14,000× *g* and 4 °C for 15 min. The supernatant was used for measuring the activity of antioxidant enzymes (SOD, POD, and CAT) and the content of MDA and H_2_O_2_.

For SOD activity, the nitro-blue tetrazolium (NBT) method was employed. The absorbance was measured using a spectrophotometer at 560 nm. The SOD activity was defined in the unit U g^−1^ FW min^−1^. For POD activity, a spectrophotometer was used to measure the absorbance at 470 nm with five replicates at an interval of 30 s, and the POD activity was expressed as U g^−1^ FW min^−1^. For CAT activity, a spectrophotometer was used to measure the absorbance at 470 nm, and the absorbance was recorded every 30 s with four replicates. The CAT activity was defined as U g^−1^ FW min^−1^. For MDA content, the absorbance was recorded by a spectrophotometer at 450, 600, and 532 nm. The MDA content was defined as µmol g^−1^ FW.

The H_2_O_2_ content was determined according to Freguson et al. (1983). Approximately 200 μL of supernatant was added to 200 μL of assay reagent (containing 500 μmol L^−1^ ammonium ferrous sulfate, 50 mmol L^−1^ H_2_SO_4_, 200 μM xylenol orange, and 200 mmol L^−1^ sorbitol). The mixture was incubated in the dark for 45 min and then measured at 560 nm. Standard curves were obtained by adding various amounts of H_2_O_2_. The H_2_O_2_ content was measured using the unit µmol g^−1^.

#### 2.3.5. Determination of Soil Chemical Properties

The rhizosphere soil was collected at 15 days after HS. The determination of soil N, P, and K contents was carried out according to Zhao et al. [36]. The soil N, P, and K contents were expressed as g kg^−1^. The soil pH and soil organic matter (SOM) were measured according to Reth et al. [37].

#### 2.3.6. Investigation of Microorganisms in the Rhizosphere

The investigation of the rhizosphere microorganisms was performed at 15 days after HS, applying FLASH v1.2.7 software. The reads of each sample were assembled according to the overlap region between two reads, resulting in Raw Tags, which were filtered using Trimmomatic v0.33 software to obtain Clean Tags. UCHIME v4.2 software was then applied to identify and remove chimeric sequences in Clean Tags, achieving the final Effective Tags. Usearch software was used to cluster Effective Tags at 97% similarity to obtain operational taxonomic units (OTUs), which were annotated by taxonomy based on the Silva (bacteria) and UNITE (fungi) taxonomy databases.

#### 2.3.7. Determination of Primary Metabolites

At 15 days after the HS, 0.5 g of leaves from the CK and S50 treatments of both varieties were taken and placed in 5 mL test tubes, then kept in liquid nitrogen, with three biological replicates each, for a total of 12 samples, and sent to Metware (Wuhan, China) for a primary metabolites analysis.

Biological samples were freeze-dried using a vacuum freeze-dryer (Scientz-100F). The freeze-dried samples were crushed using a mixer mill (400 revolutions per minute, Retsch, Germany) with a zirconia bead for 1.5 min at 30 Hz. A total of 100 mg of lyophilized powder was dissolved with 1.2 mL 70% methanol solution, vortex 30 s every 30 min for 6 times in total, and the sample was placed in a refrigerator at 4 °C overnight. Following centrifugation at 12,000 rpm for 10 min, the extracts were filtrated (SCAA-104, 0.22 μm pore size; ANPEL, Shanghai, China, http://www.anpel.com.cn/, accessed on 29 July 2020) before UPLC-MS/MS (UPLC, SHIMADZU Nexera X2, https://www.shimadzu.com.cn/, accessed on 29 July 2020; MS, Applied Biosystems 4500 Q TRAP, https://www.thermofisher.cn/cn/zh/home/brands/applied-biosystems.html accessed on 29 July 2020) analysis. A detailed description of the analysis program and data analysis is provided in Appendix A.

### 2.4. Statistical Analyses

Statistical analyses were performed using Statistix 8 (Analytical Software, Tallahassee, FL, USA). Data were analyzed by one-way analysis of variance (ANOVA) and the difference between means was estimated using the least significant difference (LSD) test at the 5% significance level. For the multivariate analysis, the data were imported into MetaboAnalyst 5.0 software [38]. Histograms and principal component analysis (PCA) were plotted using Origin 2018 software.

## 3. Results

### 3.1. Cd and Pb Accumulation and Translocation Factor

At the HS, the Cd content in the root of the two aromatic rice cultivars significantly decreased at all melatonin levels. Compared with CK, the Cd content in the shoot of Yuxiangyouzhan significantly decreased during the S50, S100, and S400 treatments. The Cd content in the leaves of the two rice cultivars decreased by 3–40% at S50, S100, and S200 (Table 1). Meanwhile, compared to CK, the S50, S100, and S200 treatments significantly reduced the TF of Cd ions from the shoot to the leaf. The TF of Cd ions from the root to the shoot was smaller than that from the shoot to the leaf (Table 2). Likewise, S100, S200, R100, and R300 treatments inhibited Pb transference from the shoot to the leaf. The S50, S100, S200, R100, and R300 treatments reduced Pb tansference from the shoot to the leaf. The TF of Pb ions from the root to the shoot was obviously smaller than that of Cd ions from the root to the shoot (Table 2). For Yuxiangyouzhan, exogenous melatonin significantly increased the enrichment of Cd ions in grains at the MS. Compared with CK, S200 treatment reduced the Cd content in grains of Xiangyaxiangzhan by 22% (Table 1). Exogenous melatonin reduced the TF of Pb ions from the stem to the leaf but increased that from the leaf to the grain (Table 2). Pb content in grains of Yuxiangyouzhan decreased by 21.37–72.92% under S50, S100, and S200 treatments, while that of Xiangyaxiangzhan significantly decreased by 9.78–43.65% at all melatonin levels of the spraying treatment and by 31.43% and 36.11% under the R100 and R300 treatments, respectively (Table 3).

### 3.2. Grain Yield and Yield-Related Traits

Spraying and irrigation with melatonin at all levels increased the grain yield of Yuxiangyouzhan, which was significantly increased by 36.13%, 45.92%, 38.24%, and 43.25% during treatments with S100, S200, R100, and R300, respectively. Additionally, S200 significantly increased the 1000-grain weight, R100 significantly increased the filled grain percentage, and R300 significantly increased the tiller number per hill and grain number per panicle. Compared to CK, the S200 and S400 treatments significantly increased the grain yield of Yuxiangyouzhan by 20.80% and 22.12%, respectively (Table 4).

### 3.3. GSH, ASA, and MTs Contents in Leaves

As the rice matured, the GSH content of the Yuxiangyouzhan leaves increased gradually. Compared with CK, the S200, S400, R300, and R500 treatments significantly increased the GSH content of the Yuxaingyouzhan leaves at the HS, while the S50, S100, S200, and S400 treatments significantly increased the GSH content at 15 days after HS and MS. In addition, R300 significantly increased the GSH content at MS, and R500 significantly increased the GSH content at 15 days after HS. For Xiangyaxiangzhan, S200 and S400 treatments increased the leaf GSH content at the HS; R300 and R500 significantly increased the leaf GSH content at 15 days after HS; and R100 significantly increased the leaf GSH content at the MS (Figure 2a,b). Compared with the HS, exogenous melatonin more obviously increased the ASA content at the MS. At the MS, the ASA content in the leaves of the two rice cultivars was promoted by the spraying and irrigation of melatonin at all concentrations. For Yuxiangyouzhan, the other treatments significantly increased the ASA content by 7.14–34.91% compared to the CK treatment. For Xiangyaxiangzhan, the ASA content increased by 17.23%, 15.39%, 24.06%, and 13.64% under S50, S100, S200, and R100 treatments, respectively (Figure 2c,d). For Yuxiangyouzhan, decreases in the MTs content in leaves were found under all treatments at the HS and MS. For Xiangyaxiangzhan, compared with CK, S400 and R100 treatments significantly increased the MTs content by 10.71% and 8.62%, respectively, at the HS; S200, S400, R300, and R500 significantly increased it by 18.72%, 17.12%, 14.66%, and 27.36% at 15 days after HS. At MS, all treatments increased the MTs content in the leaves, especially the S100, S200, R100, and R300 treatments, which remarkably increased the MTs content by 13.81%, 22.94%, 29.04%, and 38.94%, respectively (Figure 2e,f).

### 3.4. Antioxidant Enzyme Activities and MDA and H_2_O_2_ Contents in Leaves

R300 treatment increased the SOD activity, and the other concentrations of exogenous melatonin had no significant positive effect on the SOD activity (Figure 3a,b). For Yuxiangyouzhan, compared with CK, the treatments with S50, S200, and R100 significantly increased the POD activity at the HS. While S50, S100, S200, R100, R300, and R500 significantly decreased it at 15 days after HS. For Xiangyaxiangzhan, compared with CK, a significant decrease in POD activity was measured under the S100, S200, and R300 treatments. Exogenous melatonin significantly reduced the POD activity at 15 days after HS. R300 and R500 significantly increased the POD activity at MS, but other treatments showed no significant effect (Figure 3c,d). For Yuxiangyouzhan, S200 treatment significantly increased the CAT activity at the HS. S400 significantly increased the CAT activity at 15 days after HS, while other treatments decreased the CAT activity. At the MS, the S50, S400, R100, R300, and R500 treatments significantly increased the CAT activity. For Xiangyaxiangzhan, compared with CK, the treatments with S50, S100, S200, S400, R100, R300, and R500 significantly improved the CAT activity at 15 days after HS, while the S100, S200, S400, R100, R300, and R500 treatments significantly improved the CAT activity at MS. The CAT activity was the highest under R300 treatment (Figure 3e,f). Compared with CK, significant reductions in the MDA content in the Xiangyaxiangzhan leaves were found under S100 and S200 at the HS, 15 days after HS, and MS. However, for Yuxiangyouzhan, S50, S100, S200, and R500 significantly increased the MDA content at MS; exogenous melatonin treatment could not reduce the MDA content in leaves (Figure 3g,h). Compared with CK, the H_2_O_2_ content in Yuxiangyouzhan leaves decreased by 22.30–59.71% at 15 days after HS and 19.97–47.93% at the MS, respectively, under all melatonin treatments. Moreover, S50 and R300 treatments decreased the H_2_O_2_ content by 27.22% and 14.31% at the HS. For Xiangyaxiangzhan, S50 reduced the H_2_O_2_ content by 47.77% at 15 days after HS, while S50, S100, S200, R100, R300, and R500 treatments reduced the H_2_O_2_ content by 12.77–50.68% in mature leaves (Figure 3i,j).

### 3.5. Rhizosphere Microbial Community Structure

Rhizosphere microbial activities can prevent heavy metals from entering rice. The soil environment is also closely related to microorganisms. We selected the S50 and R300 treatments to investigate the microorganisms in rice rhizosphere soils. Compared to XCK, the SOM and soil N content increased, but pH decreased under XS50. The soil P content was significantly reduced under XR300. Compared to YCK, the soil N and Pb contents and pH all increased under YS50. However, the SOM and soil N content decreased under YR300 while the pH and Pb content increased (Table 5). The sparsity curve tended to be flat (Figure 4a), the species number in this environment did not increase significantly with the sequencing depth increase, indicating that the sequencing depth was sufficient, and the data were suitable for the subsequent analyses. The coverage ratio of all treatments was close to 1 (Table 6), indicating reliable results. Compared with XCK, the Shannon index under XR300 decreased significantly, while, compared with YCK, the number of OTUs under YR300 increased significantly, the ACE index increased significantly, and the Shannon index increased significantly, indicating an increase in community richness at R300 compared to the CK treatment in Yuxiangyouzhan (Table 6).

Analysis of bacterial relative abundance at the phylum level revealed the top five dominant phyla were Chloroflexi, Proteobacteria, Acidobacteria, Actinobacteria, and Gemmatimonadetes. In addition, Firmicutes, Bacteroidetes, Verrucomicrobia, Nitrospirae, and Cyanobacteria could also be detected in the soil samples (Figure 4b).

PCA showed that bacterial communities in the rice rhizospheres under S50 and R300 treatments were clearly separated from those under CK, indicating that the bacterial community structure changed under S50 and R300 treatments. Additionally, R300 appeared to be the primary factor in the first principal component axis (PC1), while S50 was the primary factor in the third principal component axis (PC3) (Figure 4c).

In addition, the redundancy analysis (RDA) plot was displayed to compare the bacterial community compositions among all soil samples and identify the major environmental variables that affected the community structure. Among all the environmental variables tested, soil Pb, Cd, P, and pH were relatively near RDA1 negative coordinates, while soil N, K, and SOM were relatively near RDA1 positive coordinates, which explained 13.36% of the total variation in the communities (Figure 4d).

The variance analysis of the top five dominant phyla showed that XR300 significantly increased the relative abundance of Actinobacteria, while significantly decreasing that of Proteobacteria compared with XCK. YR300 significantly decreased the relative abundance of Acidobacteria while significantly increasing that of Proteobacteria; there was no significant difference in the relative abundance of each dominant phylum between XS50 and YS50 (Table 7). Besides, there were significant differences in the community structures of rhizosphere bacteria between the two rice cultivars. For Xiangyaxiangzhan, the relative abundance of Proteobacteria under XR300 was significantly lower than that under XCK and XS50; however, the opposite was true for Yuxiangyouzhan (Table 7).

### 3.6. Investigation of Primary Metabolites

#### 3.6.1. Metabolite Profiling

Leaf metabolites under S50 at 15 days after HS for each cultivar were investigated, and a total of 395 metabolites was detected. The primary attribution of these metabolites consisted of 79 amino acids and derivatives, 86 phenolic acids, 40 nucleotides and derivatives, and 71 organic acids and lipids; additionally, 47 other metabolites such as vitamins, saccharides, and alcohol were also detected (Appendix A).

In general, the metabolic pathways such as ‘alanine, aspartate, and glutamate metabolism’, ‘arginine biosynthesis’, ‘aminoacyl-tRNA biosynthesis’, ‘nicotinate and nicotinamide metabolism’, ‘phenylpropanoid biosynthesis’, ‘citrate cycle (TCA cycle)’, ‘butanoate metabolism’, ‘arginine and proline metabolism’, ‘C5-branched dibasic acid metabolism’, ‘valine, leucine and isoleucine biosynthesis’, ‘pantothenate and CoA biosynthesis’, ‘glyoxylate and dicarboxylate metabolism’, ‘carbon fixation in photosynthetic organisms’, ‘pyrimidine metabolism’, ‘isoquinoline alkaloid biosynthesis’, ‘starch and sucrose metabolism’, ‘beta-alanine metabolism’, ‘glycine, serine and threonine metabolism’, ‘vitamin B6 metabolism’, and ‘betalain biosynthesis’ were enriched among the detected metabolites (Figure 5).

PCA and PLS-DA (Partial Least Squares-Discriminant Analysis) were performed to analyze the metabolic profiles of the leaves of the two cultivars under all experimental treatments (Figure 6a,b). More than 90% of the variation could be explained by PC1 and PC2. Therefore, 31 metabolites with VIP scores ˃ 1 were considered the core compounds to distinguish the treatments (Figure 6c). Among these core metabolites, 6, 19, 2, and 4 metabolites showed the highest values under XCK, XS50, YCK, and YS50, respectively (Figure 6c). The PCA result showed that the leaf metabolomes under XCK and XS50 were separated completely along PC1, which accounted for 94.3% of the total variation. The dominant metabolites (VIP scores > 4) contributing to PC1 were mws2623 (vaccenic acid), pme3961 (2′-deoxyadenosine), mws1491 (linoleic acid), pmb0855 (lysoPC 16:0), mws0250 (L-tyrosine), and mws1489 (stearic Acid) (Figure 6e,f). A total of 31 metabolites (VIP scores > 4) under YCK and YS50 treatments were separated completely along PC1, which accounted for 54.6% of the total variation. The dominant metabolites contributing to PC1 were mws0192 (succinic acid), mws0256 (L-valine), mws0281 (citric acid), mws0258 (L-isoleucine), mws0227 (L-leucine), and pmc0066 (2′-deoxyinosine-5′-monophosphate) (Figure 6h,i).

#### 3.6.2. Differential Metabolites

A Venn diagram was constructed to show the differential metabolites identified in leaves at 15 days after HS for each cultivar. There were 54, 36, 47, and 32 significant differential metabolites identified for XCK vs. XS50, XCK vs. YCK, XS50 vs. YS50, and YCK vs. YS50, respectively, which included 6, 18, 40, and 19 down-regulated metabolites and 48, 18, 7, and 13 up-regulated metabolites, respectively (Figure 7).

## 4. Discussion

When heavy metals are present in the soil environment, Cd and Pb enter rice from the roots, and, therefore, Cd and Pb accumulate first in the roots [39]. Once excessive heavy metals are accumulated, these ions are transported upward through transpiration, and finally accumulate in the shoot, leaf, and grain, causing heavy metal stress during rice growth [40]. It is crucial to reduce the Cd and Pb content in rice grains and Cd toxicity to plants. Hence, we investigated the effect of melatonin on Cd and Pb content in rice grains, grain yield and antioxidant capacity, soil chemistry, microbial structure, and related metabolism to investigate the mechanisms of melatonin regulation on the mitigation of heavy metal stress in aromatic rice.

Ashraf et al. [4] reported that the key to reducing Pb content in rice grains was to reduce the TF of Pb from the stem to the leaf in rice. Our results show that the administration of melatonin had a significant effect on reducing the Pb content in rice grains. Reduced Pb content in grains was found in both aromatic rice cultivars under spraying treatments (Table 3). In line with our results, Posmyk et al. [41] reported that pre-sowing seed treatment with melatonin protected red cabbage seedlings from copper toxicity. Many experiments have demonstrated that melatonin can alleviate heavy metal stress in plants.

GSH is a precursor of plant chelating peptides and can be synthesized with Cd^2+^ under GSH thiotransferase catalysis [42,43]. MTs can directly chelate Pb and Cd. Therefore, high GSH and MTs contents have a positive effect on alleviating Cd–Pb toxicity. Furthermore, the GSH-ASA cycle is one of the pathways for scavenging ROS [44]. Our results show that the spraying of melatonin markedly increased the GSH content at the MS (Figure 2), which was consistent with a previous report that melatonin regulated the biosynthesis of GSH in response to Cd–Pb stress [45]. Under S100, S200, R100, and R300 treatments, there was a significant increase in the MTs content of Xiangyaxiangzhan leaves, with reduced the Pb content observed in the grains, which may be induced by exogenous melatonin application (Figure 2 and Table 3).

Additionally, Cd and Pb stress induces the accumulation of intracellular ROS [46,47]. The production of ROS and the imbalance of antioxidant capacity are among the causes of heavy metal toxicity [48,49]. Our results show that S50, S100, S200, R100, R300, and R500 treatments significantly reduced the accumulation of H_2_O_2_ in rice leaves at the MS under Cd–Pb stress (Figure 3), resulting in reduced MDA content in Xiangyaxiangzhan leaves (Figure 3). This result is in line with Zhang et al. [50], which showed that adding melatonin decreased the MDA concentration in cucumber. This suggests that melatonin can increase the antioxidant system and improve the antioxidant damage capacity of the plant.

Melatonin not only directly scavenges ROS [51,52], but also induces the activity of the antioxidant enzymes and optimizes the transfer of electrons through the electron transport chain of the inner mitochondrial membrane [53,54]. Hosseinzadeh et al. [55] expressed that the indirect antioxidant activity of melatonin involves increasing mRNA levels and the SOD activity of the membrane. In contrast, our study shows that exogenous melatonin application had no significant positive effect on SOD activity, while R100, R300, and R500 significantly improved the CAT activity at the MS in both rice cultivars (Figure 3). Unfortunately, the endogenous melatonin content was not measured in our study, but the application of exogenous melatonin positively affected the rice physiological growth from the above.

Microbial activities in the soil rhizosphere soil may reduce the uptake of heavy metals by plants. The higher the relative abundance of soil rhizosphere microorganisms, the lower the uptake of heavy metals. In this study, YR300 significantly increased the OTU number, ACE index, and Shannon index, indicating that the relative abundance of soil bacteria was higher under YR300 than YCK. Meanwhile, XR300 significantly increased the Simpson index. Our experiments show that high concentrations of melatonin applied to the roots cause changes in the soil’s microbial structure. RDA indicated that the soil Cd content plays an important role in altering bacterial community structures (Table 6).

Melatonin, as a plant-growth regulator, can improve plant quality [56,57]. Pyruvate is an important intermediate during the glucose metabolism of all cells and mutually transforms various substances in vivo. Pyruvate is a substrate of the TCA cycle, and increased pyruvate content is beneficial to the normal development of the TCA cycle under Cd–Pb stress. Oxaloacetate reduces the bioavailability of mineral elements and is considered as an antagonist of mineral element absorption and utilization [58]. Proline and arginine are indispensable osmotic-adjusting substances [59]. Studies have shown that ZnO nanomaterial alleviates cadmium poisoning by changing the proline and arginine content of aromatic rice [60]. In our experiment, metabolites were significantly altered and increased in both varieties under the S50 treatment compared to the CK treatment (Figure 6 and Figure 7). Increased amino acid content and quantity is beneficial to maintaining high osmotic pressure in the cell and preventing the entry of Cd and Pb into the cell.

## 5. Conclusions

The S50, S100, and S200 treatments alleviated the symptoms caused by Cd–Pb toxicity in rice, as evidenced by the decreased Pb accumulation in the grain and the increased grain yield. It can be seen that spraying treatments are more effective in reducing Cd–Pb content levels in aromatic rice. Furthermore, S50 and R300 exhibited the most significant effect on reducing Pb content in grains, and both rice cultivars showed increased GSH, ASA, and MTs content and a decreased H_2_O_2_ content in leaves at the MS. Meanwhile, S50 and R300 changed the rhizosphere bacterial structures, even though no marked difference was observed in bacterial diversity. It can be seen that 50 μmol L^−1^ for the spraying treatment and 300 μmol L^−1^ for the irrigation treatment are suitable concentrations for melatonin application to aromatic rice. Melatonin application increased soil N and K contents, decreased soil pH, and changed soil physicochemical properties, which probably played the most critical role in altering the bacterial community structure to prevent heavy metal toxicity. Besides, S50 promoted the synthesis of various amino acids; alanine, aspartate, and glutamate metabolism; and arginine biosynthesis. This demonstrates that spraying low concentrations of melatonin in the leaves promotes the production of metabolites in aromatic rice, which was possibly closely associated with the antioxidant capacity of rice to heavy metal stress.

## Figures and Tables

**Figure 1 antioxidants-11-00776-f001:**
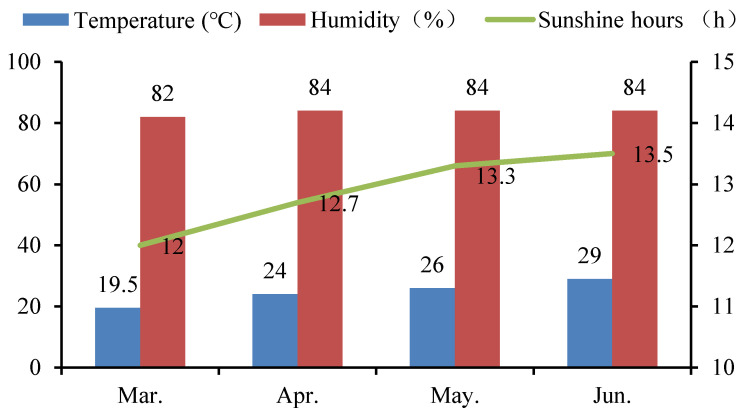
The average monthly temperature, humidity, and sunshine hours during the experiment period in 2019.

**Figure 2 antioxidants-11-00776-f002:**
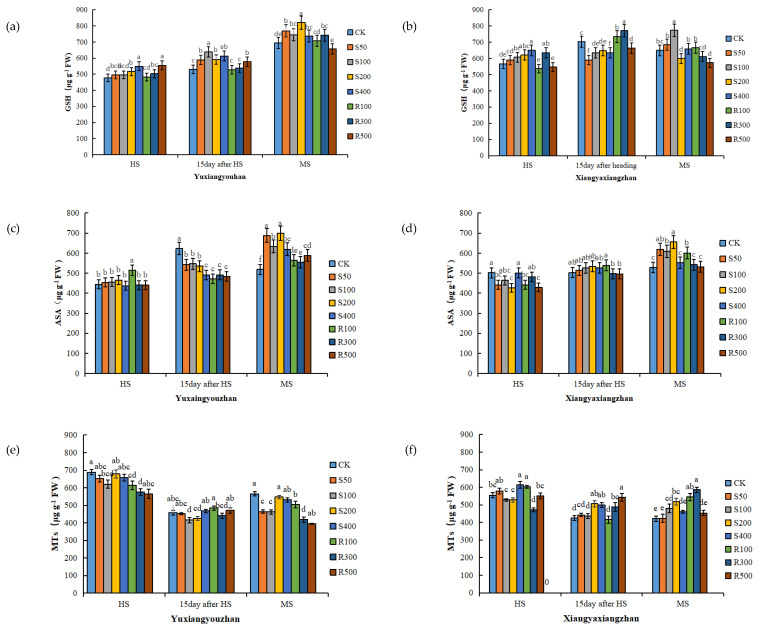
Effects of exogenous melatonin application on GSH content in Yuxiangyouzhan (**a**) and Xiangyaxiangzhan (**b**); ASA content in Yuxiangyouzhan (**c**) and Xiangyaxiangzhan (**d**); MTs content in Yuxiangyouzhan (**e**) and Xiangyaxiangzhan (**f**) at the heading stage (HS), 15 day after HS, and the maturity stage (MS). Means sharing similar letters indicate no significant difference at *p* < 0.05 according to the LSD test.

**Figure 3 antioxidants-11-00776-f003:**
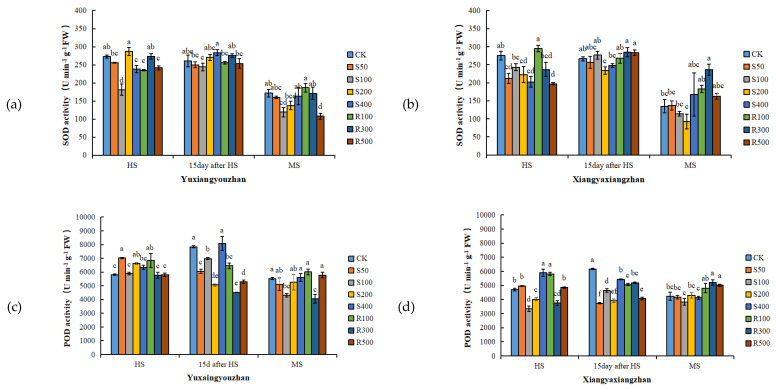
Effects of exogenous melatonin application on SOD activity in Yuxiangyouzhanand (**a**) and Xiangyaxiangzhan (**b**); POD activity in Yuxiangyouzhan (**c**) and Xiangyaxiangzhan (**d**); CAT activity in Yuxiangyouzhan (**e**) and Xiangyaxiangzhan (**f**); MDA content in Yuxiangyouzhan (**g**) and Xiangyaxiangzhan (**h**); and H_2_O_2_ content in Yuxiangyouzhan (**i**) and Xiangyaxiangzhan (**j**) at the heading stage (HS), 15 day after HS, and the MS. Means sharing similar letters indicate no significant difference at *p* < 0.05 according to the LSD test.

**Figure 4 antioxidants-11-00776-f004:**
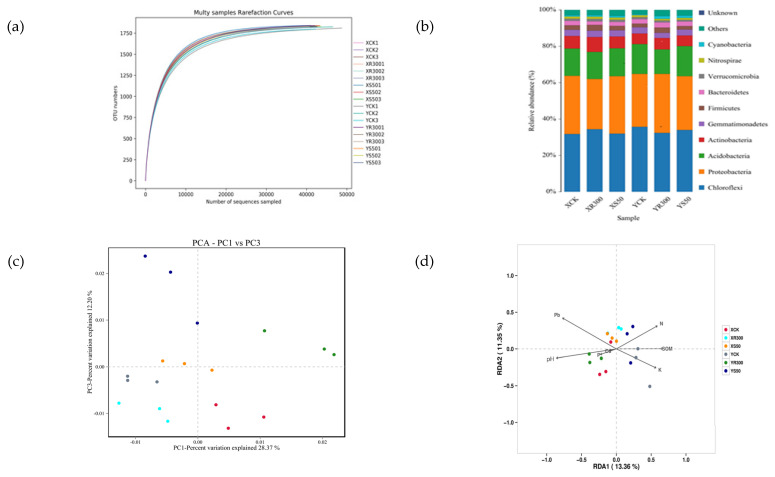
Rarefaction curve (**a**); histogram of species distribution (**b**); PCA plot of fungal communities in 18 soil samples under the XCK, XS50, YCK, YR300 treatments (**c**); RDA considering the fungal relative abundance at the operational taxonomic unit (OTU) level; SOM, soil total N, P, and K contents, Cd and Pb concentrations, and pH (**d**). XCK: Xiangyaxiangzhan growing under Cd–Pb stress without melatonin treatment; XR300: Xiangyaxiangzhan growing under Cd–Pb stress and irrigated with 300 μmol L^−1^ exogenous melatonin; XS50: Xiangyaxiangzhan growing under Cd–Pb stress and sprayed with 50 μmol L^−1^ exogenous melatonin; YCK: Yuxiangyouzhan growing under Cd–Pb stress without melatonin treatment; YR300: Yuxiangyouzhan growing under Cd–Pb stress and irrigated with 300 μmol L^−1^ exogenous melatonin; YS50: Yuxiangyouzhan growing under Cd–Pb stress and sprayed with 50 μmol L^−1^ exogenous melatonin.

**Figure 5 antioxidants-11-00776-f005:**
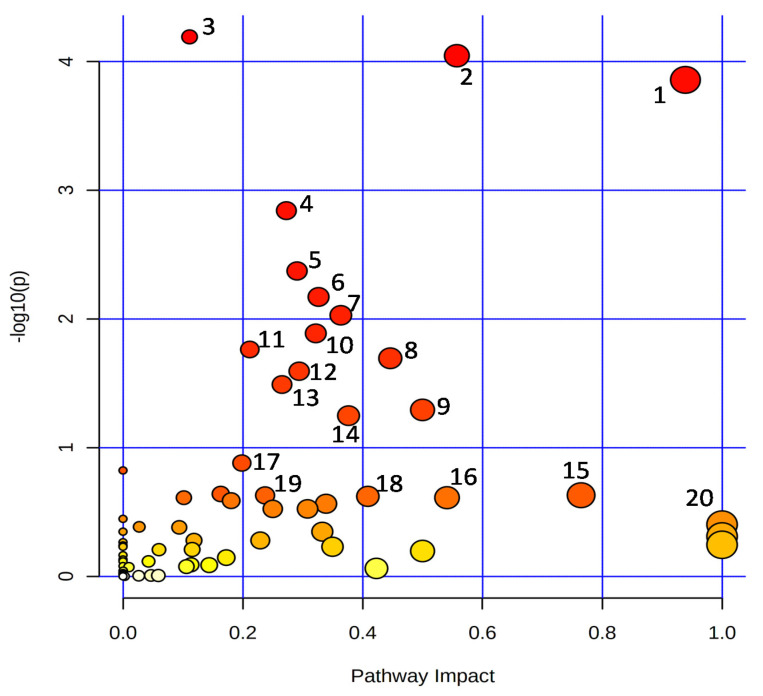
Overview of the enriched KEGG pathways. 1: Alanine, aspartate, and glutamate metabolism; 2: Arginine biosynthesis; 3: Aminoacyl-tRNA biosynthesis; 4: Nicotinate and nicotinamide metabolism; 5: Phenylpropanoid biosynthesis; 6: Citrate cycle (TCA cycle); 7: Butanoate metabolism; 8: Arginine and proline metabolism; 9: C5-Branched dibasic acid metabolism; 10: Valine, leucine, and isoleucine biosynthesis; 11: Pantothenate and CoA biosynthesis; 12: Glyoxylate and dicarboxylate metabolism; 13: Carbon fixation in photosynthetic organisms; 14: Pyrimidine metabolism; 15: Isoquinoline alkaloid biosynthesis; 16: Starch and sucrose metabolism; 17: beta-Alanine metabolism; 18: Glycine, serine, and threonine metabolism; 19: Vitamin B6 metabolism; 20: Betalain biosynthesis.

**Figure 6 antioxidants-11-00776-f006:**
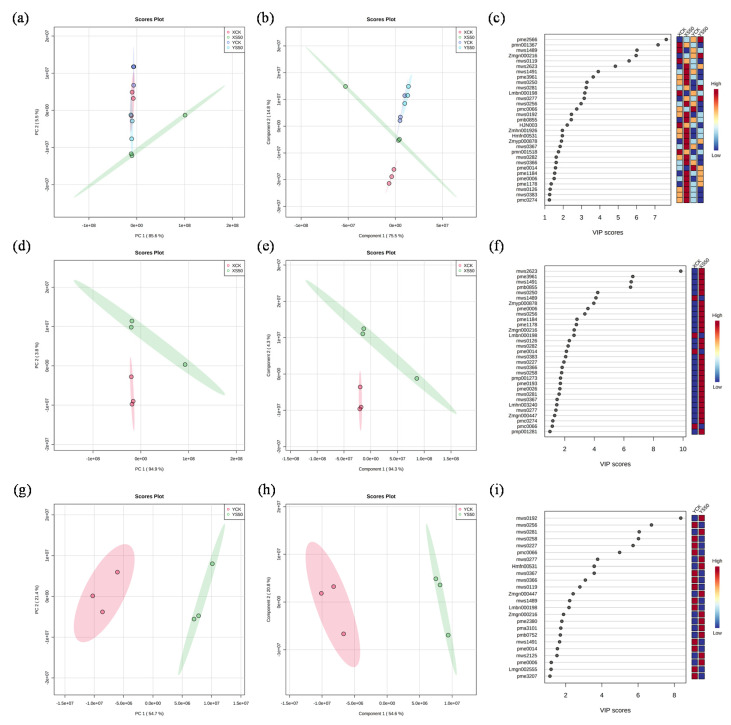
Multivariate pattern recognition analysis of metabolomics. (**a**) PCA plot of metabolites under XCK, XS50, YCK, and YS50 treatments; (**b**) PLS-DA plot of metabolites under XCK, XS50, YCK, and YS50 treatments; (**c**) VIP score of metabolites under XCK, XS50, YCK, and YS50 treatments; (**d**) PCA plot of metabolites under XCK and XS50 treatments; (**e**) PLS-DA plot of metabolites under XCK and XS50 treatments; (**f**) VIP score of metabolites under XCK and XS50 treatments; (**g**) PCA plot of metabolites under YCK and YS50 treatments; (**h**) PLS-DA plot of metabolites under YCK and YS50 treatments; and (**i**) VIP score of metabolites under YCK and YS50 treatments.

**Figure 7 antioxidants-11-00776-f007:**
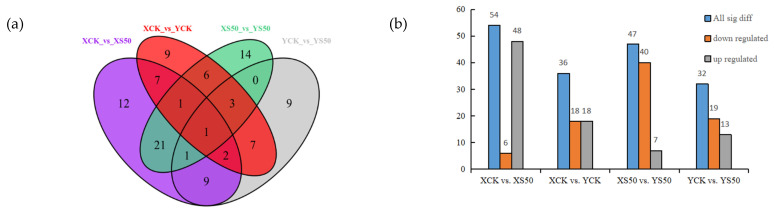
(**a**) Venn diagram showing the numbers of differential metabolites in XCK vs. XS50, XCK vs. YCK, XS50 vs. YS50, and YCK vs. YS50 comparisons. (**b**) Significant differential metabolites in XCK vs. XS50, XCK vs. YCK, XS50 vs. YS50, and YCK vs. YS50.

**Table 1 antioxidants-11-00776-t001:** Effects of exogenous melatonin application on Cd content in different plant parts (mg kg^−1^).

	Treatments	Cd Content at the HS (mg kg^−1^)			Cd Content at MS (mg kg^−1^)			
		Roots	Shoots	Leaves	Roots	Shoots	Leaves	Grains
Yuxiangyouzhan	CK	0.6959 ± 0.0097 ^a^	0.3190 ± 0.0029 ^b^	0.2452 ± 0.0056 ^b^	0.6351 ± 0.0088 ^c^	0.4449 ± 0.0117 ^e^	0.3781 ± 0.0412 ^cd^	0.1101 ± 0.0019 ^f^
	S50	0.6391 ± 0.0067 ^b^	0.2964 ± 0.0116 ^c^	0.2174 ± 0.0041 ^bc^	0.5544 ± 0.0227 ^c^	0.5452 ± 0.0099 ^b^	0.3882 ± 0.0026 ^cd^	0.1404 ± 0.0047 ^e^
	S100	0.6337 ± 0.0023 ^b^	0.2946 ± 0.0058 ^c^	0.1542 ± 0.0199 ^d^	0.8774 ± 0.0384 ^b^	0.4299 ± 0.0037 ^e^	0.350 ± 0.0039 ^cd^	0.1444 ± 0.0012 ^de^
	S200	0.6520 ± 0.0005 ^b^	0.3284 ± 0.0006 ^ab^	0.1920 ± 0.0041 ^c^	0.6587 ± 0.0074 ^c^	0.5146 ± 0.0038 ^cd^	0.4646 ± 0.0141 ^a^	0.1539 ± 0.0034 ^c^
	S400	0.6352 ± 0.0123 ^b^	0.2964 ± 0.0015 ^c^	0.2942 ± 0.0174 ^a^	1.0646 ± 0.107 ^a^	0.5317 ± 0.0097 ^bc^	0.3724 ± 0.0146 ^cd^	0.1512 ± 0.0008 ^cd^
	R100	0.6532 ± 0.0085 ^b^	0.345 ± 0.0122 ^b^	0.3203 ± 0.0026 ^a^	0.9645 ± 0.0402 ^ab^	0.6179 ± 0.0147 ^a^	0.4501 ± 0.0042 ^ab^	0.1754 ± 0.0015 ^b^
	R300	0.6398 ± 0.0039 ^b^	0.3264 ± 0.0051 ^ab^	0.2473 ± 0.0109 ^b^	0.9879 ± 0.019 ^ab^	0.5421 ± 0.0017 ^bc^	0.3395 ± 0.0052 ^d^	0.1687 ± 0.0006 ^b^
	R500	0.6541 ± 0.0032 ^b^	0.3463 ± 0.005 ^b^	0.3234 ± 0.0038 ^a^	1.0074 ± 0.0199 ^ab^	0.4944 ± 0.0105 ^d^	0.3993 ± 0.0055 ^bc^	0.1844 ± 0.0017 ^a^
Xiangyaxiangzhan	CK	0.7083 ± 0.0054 ^a^	0.3518 ± 0.0063 ^d^	0.3285 ± 0.0025 ^bc^	1.0652 ± 0.0135 ^b^	0.5453 ± 0.0114 ^a^	0.4977 ± 0.005 ^a^	0.1753 ± 0.0012 ^e^
	S50	0.6390 ± 0.0018 ^d^	0.3251 ± 0.0002 ^g^	0.2756 ± 0.0062 ^e^	1.0205 ± 0.0105 ^c^	0.5003 ± 0.0047 ^b^	0.4131 ± 0.002 ^d^	0.1919 ± 0.0002 ^b^
	S100	0.6308 ± 0.0002 ^d^	0.3470 ± 0.0013 ^de^	0.3202 ± 0.0015 ^bcd^	1.1254 ± 0.0073 ^a^	0.4860 ± 0.0076 ^b^	0.4583 ± 0.0167 ^bc^	0.1749 ± 0.0001 ^e^
	S200	0.5923 ± 0.0014 ^e^	0.3631 ± 0.0007 ^c^	0.2985 ± 0.0029 ^de^	0.9774 ± 0.0073 ^d^	0.3441 ± 0.0025 ^c^	0.3596 ± 0.0065 ^e^	0.1431 ± 0.0017 ^f^
	S400	0.6608 ± 0.0012 ^c^	0.3957 ± 0.004 ^b^	0.3596 ± 0.0029 ^a^	1.0145 ± 0.0024 ^c^	0.4987 ± 0.0013 ^b^	0.4249 ± 0.0037 ^cd^	0.1858 ± 0.0011 ^c^
	R100	0.6383 ± 0.0056 ^d^	0.3329 ± 0.0021 ^fg^	0.3027 ± 0.0007 ^cde^	1.0829 ± 0.0167 ^b^	0.4973 ± 0.0045 ^b^	0.4029 ± 0.0049 ^bc^	0.1813 ± 0.0005 ^d^
	R300	0.6782 ± 0.0092 ^b^	0.3377 ± 0.0026 ^ef^	0.2815 ± 0.0265 ^e^	1.0676 ± 0.005 ^b^	0.536 ± 0.0169 ^a^	0.4386 ± 0.0063 ^bc^	0.1758 ± 0.0013 ^e^
	R500	0.5869 ± 0.0027 ^e^	0.4063 ± 0.0059 ^a^	0.3335 ± 0.0072 ^ab^	1.0186 ± 0.0149 ^c^	0.4874 ± 0.0074 ^b^	0.4189 ± 0.0078 ^cd^	0.2074 ± 0.0002 ^c^

HS: heading stage; MS: maturity stage. CK: rice growing under Cd–Pb stress without melatonin treatment; S50: rice growing under Cd–Pb stress and sprayed with 50 μmol L^−1^ exogenous melatonin; S100: rice growing under Cd–Pb stress and sprayed with 100 μmol L^−1^ exogenous melatonin; S200: rice growing under Cd–Pb stress and sprayed with 200 μmol L^−1^ exogenous melatonin; S400: rice growing under Cd–Pb stress and sprayed with 400 μmol L^−1^ exogenous melatonin; R100: rice growing under Cd–Pb stress and irrigated with 100 μmol L^−1^ exogenous melatonin; R300: rice growing under Cd–Pb stress and irrigated with 300 μmol L^−1^ exogenous melatonin; R500: rice growing under Cd–Pb stress and irrigated with 500 μmol L^−1^ exogenous melatonin. Means sharing same letters indicate no significant difference at *p* < 0.05 according to the LSD test.

**Table 2 antioxidants-11-00776-t002:** Effects of exogenous melatonin application on the translocation factor of Cd and Pb.

Cultivars	Treatments	TF of HS(Cd)		TF of MS(Cd)			TF of HS(Pb)		TF of MS(Pb)		
		R to S	S to L	R to S	S to L	L to G	R to S	S to L	R to S	S to L	L to G
Yuxiangyouzhan	CK	0.4583	0.7687	0.7005	0.8500	0.2912	0.1631	0.7646	0.1177	0.9260	0.0487
	S50	0.4637	0.7335	0.9835	0.7120	0.3617	0.1849	0.6960	0.1986	0.6255	0.0122
	S100	0.4648	0.5236	0.4899	0.8142	0.4127	0.1879	0.5988	0.1166	0.7683	0.0411
	S200	0.5036	0.5846	0.7812	0.9029	0.3312	0.1456	0.5175	0.2066	0.5529	0.0299
	S400	0.4666	0.9927	0.4994	0.7004	0.4060	0.1945	0.6210	0.1505	0.6794	0.0473
	R100	0.5282	0.9283	0.6406	0.7284	0.3898	0.1478	0.7407	0.1705	0.9445	0.0254
	R300	0.5101	0.7578	0.5488	0.6262	0.4969	0.1928	0.4544	0.1167	0.8652	0.0539
	R500	0.5294	0.9339	0.4908	0.8076	0.4617	0.1880	0.9723	0.1420	0.4224	0.1423
Xiangyaxiangzhan	CK	0.4967	0.9337	0.5119	0.9127	0.3523	0.1523	0.7218	0.1426	0.7639	0.0844
	S50	0.5088	0.8478	0.4903	0.8256	0.4646	0.1469	0.6150	0.1542	0.6247	0.0729
	S100	0.5501	0.9228	0.4319	0.9429	0.3817	0.1660	0.5222	0.1978	0.4477	0.1010
	S200	0.6130	0.8220	0.3521	1.0450	0.3980	0.1630	0.6848	0.1514	0.6047	0.1251
	S400	0.5988	0.9088	0.4916	0.8520	0.4373	0.1459	0.8065	0.2444	0.7118	0.0517
	R100	0.5215	0.9095	0.4593	0.8102	0.4499	0.1293	0.7340	0.1474	0.6758	0.0644
	R300	0.4980	0.8334	0.5021	0.8183	0.4007	0.1123	0.7577	0.1613	0.5962	0.0613
	R500	0.6922	0.8209	0.4785	0.8595	0.4950	0.2372	0.8762	0.1646	0.7682	0.0972

TF: translocation factor. HS: heading stage; MS: maturity stage. CK: rice growing under Cd–Pb stress without melatonin treatment; S50: rice growing under Cd–Pb stress and sprayed with 50 μmol L^−1^ exogenous melatonin; S100: rice growing under Cd–Pb stress and sprayed with 100 μmol L^−1^ exogenous melatonin; S200: rice growing under Cd–Pb stress and sprayed with 200 μmol L^−1^ exogenous melatonin; S400: rice growing under Cd–Pb stress and sprayed with 400 μmol L^−1^ exogenous melatonin; R100: rice growing under Cd–Pb stress and irrigated with 100 μmol L^−1^ exogenous melatonin; R300: rice growing under Cd–Pb stress and irrigated with 300 μmol L^−1^ exogenous melatonin; R500: rice growing under Cd–Pb stress and irrigated with 500 μmol L^−1^ exogenous melatonin; R to S: root to shoot translocation factor; S to L: shoot to leaf translocation factor; L to G: shoot to grain translocation factor.

**Table 3 antioxidants-11-00776-t003:** Effects of exogenous melatonin application on Pb content in different plant parts (mg kg^−1^).

	Treatments	Pb Content at the HS (mg kg^−1^)			Pb Content at MS (mg kg^−1^)			
		Roots	Shoots	Leaves	Roots	Shoots	Leaves	Grains
Yuxiangyouzhan	CK	2.2953 ± 0.0317 ^b^	0.3744 ± 0.0031 ^ab^	0.2863 ± 0.0026 ^b^	2.3747 ± 0.0126 ^ef^	0.2795 ± 0.0086 ^c^	0.2588 ± 0.0179 ^d^	0.0126 ± 0.0002 ^cd^
	S50	2.2107 ± 0.0046 ^c^	0.4087 ± 0.0103 ^ab^	0.2844 ± 0.0021 ^b^	2.2534 ± 0.0156 ^f^	0.4474 ± 0.0160 ^b^	0.2799 ± 0.0094 ^cd^	0.0034 ± 0.0004 ^f^
	S100	2.2429 ± 0.0091 ^c^	0.4215 ± 0.0133 ^ab^	0.2524 ± 0.0059 ^c^	2.6869 ± 0.0284 ^d^	0.3133 ± 0.0081 ^c^	0.2407 ± 0.0058 ^de^	0.0099 ± 0.0008 ^de^
	S200	2.3688 ± 0.0052 ^a^	0.3448 ± 0.0278 ^b^	0.1784 ± 0.0017 ^e^	2.6092 ± 0.027 ^de^	0.5392 ± 0.0284 ^a^	0.2981 ± 0.0186 ^cd^	0.0089 ± 0.0006 ^e^
	S400	2.3190 ± 0.0067 ^b^	0.4511 ± 0.0657 ^a^	0.2801 ± 0.0006 ^b^	3.1400 ± 0.1116 ^c^	0.4726 ± 0.0235 ^b^	0.3211 ± 0.0049 ^bc^	0.0152 ± 0.0015 ^c^
	R100	2.3833 ± 0.0167 ^a^	0.3522 ± 0.0051 ^b^	0.2609 ± 0.0095 ^c^	3.4200 ± 0.0556 ^ab^	0.5832 ± 0.0374 ^a^	0.5508 ± 0.0305 ^a^	0.0140 ± 0.0014 ^c^
	R300	2.3221 ± 0.0059 ^b^	0.4476 ± 0.0397 ^a^	0.2034 ± 0.0035 ^d^	3.5829 ± 0.0932 ^a^	0.4182 ± 0.0103 ^b^	0.3618 ± 0.0179 ^b^	0.0195 ± 0.0018 ^b^
	R500	2.3900 ± 0.0080 ^a^	0.4493 ± 0.0018 ^a^	0.4369 ± 0.0051 ^a^	3.1836 ± 0.166 ^bc^	0.4521 ± 0.0167 ^b^	0.1909 ± 0.0092 ^e^	0.0272 ± 0.0011 ^a^
Xiangyaxiangzhan	CK	2.3847 ± 0.0035 ^bc^	0.3632 ± 0.0169 ^bc^	0.2621 ± 0.004 ^bc^	2.7164 ± 0.0263 ^a^	0.3873 ± 0.0378 ^b^	0.2958 ± 0.0125 ^b^	0.0250 ± 0.0012 ^b^
	S50	2.2789 ± 0.0065 ^e^	0.3348 ± 0.0211 ^bc^	0.2059 ± 0.0197 ^d^	2.0039 ± 0.1730 ^c^	0.3090 ± 0.0127 ^c^	0.1930 ± 0.0018 ^d^	0.0141 ± 0.0007 ^e^
	S100	2.3133 ± 0.0020 ^de^	0.3839 ± 0.0017 ^b^	0.2005 ± 0.0071 ^d^	2.3002 ± 0.0713 ^b^	0.4588 ± 0.0141 ^a^	0.2054 ± 0.0105 ^d^	0.0207 ± 0.0005 ^bcd^
	S200	2.2784 ± 0.0192 ^e^	0.3713 ± 0.0096 ^b^	0.2543 ± 0.0058 ^c^	1.9672 ± 0.0902 ^c^	0.2978 ± 0.0224 ^c^	0.1800 ± 0.0037 ^d^	0.0225 ± 0.0004 ^bc^
	S400	2.4662 ± 0.0129 ^a^	0.3597 ± 0.0434 ^bc^	0.2901 ± 0.0056 ^b^	1.9096 ± 0.0877 ^c^	0.4666 ± 0.0066 ^a^	0.3321 ± 0.0075 ^a^	0.0172 ± 0.0013 ^cde^
	R100	2.3322 ± 0.0149 ^cd^	0.3016 ± 0.0208 ^cd^	0.2214 ± 0.0048 ^d^	2.6686 ± 0.0395 ^a^	0.3933 ± 0.0135 ^b^	0.2658 ± 0.0033 ^bc^	0.0171 ± 0.0005 ^cde^
	R300	2.4035 ± 0.0423 ^b^	0.2698 ± 0.0077 ^d^	0.2044 ± 0.0027 ^d^	2.7063 ± 0.1484 ^a^	0.4365 ± 0.0046 ^ab^	0.2602 ± 0.0043 ^cd^	0.0160 ± 0.0017 ^de^
	R500	2.3678 ± 0.0034 ^bc^	0.5616 ± 0.0191 ^a^	0.4921 ± 0.0208 ^a^	2.8137 ± 0.0294 ^a^	0.4632 ± 0.004 ^a^	0.3559 ± 0.0265 ^a^	0.0346 ± 0.0057 ^a^

HS: heading stage; MS: maturity stage. CK: rice growing under Cd–Pb stress without melatonin treatment; S50: rice growing under Cd–Pb stress and sprayed with 50 μmol L^−1^ exogenous melatonin; S100: rice growing under Cd–Pb stress and sprayed with 100 μmol L^−1^ exogenous melatonin; S200: rice growing under Cd–Pb stress and sprayed with 200 μmol L^−1^ exogenous melatonin; S400: rice growing under Cd–Pb stress and sprayed with 400 μmol L^−1^ exogenous melatonin; R100: rice growing under Cd–Pb stress and irrigated with 100 μmol L^−1^ exogenous melatonin; R300: rice growing under Cd–Pb stress and irrigated 300 μmol L^−1^ exogenous melatonin; R500: rice growing under Cd–Pb stress and irrigated with 500 μmol L^−1^ exogenous melatonin. Means sharing same letters indicate no significant difference at *p* < 0.05 according to the LSD test.

**Table 4 antioxidants-11-00776-t004:** Effects of exogenous melatonin application on grain yield and yield components.

Cultivars	Treatment	Tiller Number Per Hill	Grain Number Per Panicle	Filled Grain Percentage (%)	1000-Grain Weight (g)	Grain Yield (g/Pot)
Yuxiangyouzhan	CK	6.42 ± 0.21 ^b^	117.05 ± 6.18 ^b^	65.01 ± 4.64 ^b^	19.28 ± 0.2 ^b^	45.34 ± 0.99 ^c^
	S50	7.53 ± 0.35 ^a^	122.81 ± 16.13 ^ab^	66.14 ± 3.75 ^b^	19.75 ± 0.23 ^ab^	52.32 ± 2.81 ^bc^
	S100	6.89 ± 0.53 ^ab^	133.07 ± 3.61 ^ab^	71.37 ± 4.81 ^ab^	19.64 ± 0.25 ^ab^	61.72 ± 2.26 ^ab^
	S200	6.83 ± 0.17 ^ab^	126.98 ± 2.02 ^ab^	75.73 ± 3.55 ^ab^	20.13 ± 0.32 ^a^	66.16 ± 3.68 ^a^
	S400	6.46 ± 0.18 ^b^	120.24 ± 6.29 ^b^	68.35 ± 2.43 ^ab^	19.70 ± 0.33 ^ab^	49.92 ± 3.31 ^c^
	R100	6.67 ± 0.14 ^ab^	134.52 ± 8.84 ^ab^	79.83 ± 0.59 ^a^	19.25 ± 0.25 ^b^	62.68 ± 5.13 ^ab^
	R300	7.50 ± 0.40 ^a^	144.34 ± 8.41 ^a^	68.94 ± 5.20 ^ab^	18.97 ± 0.20 ^b^	64.95 ± 3.09 ^a^
	R500	6.78 ± 0.30 ^ab^	128.03 ± 8.33 ^ab^	68.87 ± 4.07 ^ab^	19.32 ± 0.25 ^b^	53.86 ± 5.47 ^bc^
Xiangyaxiangzhan	CK	7.48 ± 0.29 ^b^	75.58 ± 10.14 ^b^	80.78 ± 4.56 ^a^	19.30 ± 0.12 ^a^	44.84 ± 1.41 ^b^
	S50	8.00 ± 0.43 ^ab^	83.02 ± 6.32 ^ab^	80.92 ± 1.29 ^a^	19.05 ± 0.25 ^ab^	47.07 ± 1.96 ^b^
	S100	7.50 ± 0.18 ^b^	81.59 ± 2.60 ^ab^	82.76 ± 0.64 ^a^	19.37 ± 0.35 ^a^	47.17 ± 1.17 ^b^
	S200	8.69 ± 0.47 ^a^	81.04 ± 7.60 ^ab^	82.36 ± 2.27 ^a^	18.91 ± 0.49 ^ab^	54.18 ± 2.26 ^a^
	S400	7.58 ± 0.75 ^ab^	97.10 ± 6.73 ^a^	76.60 ± 2.47 ^a^	18.96 ± 0.47 ^ab^	54.76 ± 1.90 ^a^
	R100	8.08 ± 0.55 ^ab^	89.12 ± 2.81 ^ab^	78.77 ± 2.40 ^a^	18.09 ± 0.26 ^b^	49.38 ± 0.80 ^ab^
	R300	7.69 ± 0.18 ^ab^	96.13 ± 3.23 ^a^	76.83 ± 5.89 ^a^	18.46 ± 0.19 ^ab^	49.55 ± 1.82 ^ab^
	R500	7.42 ± 0.25 ^b^	81.15 ± 7.56 ^ab^	76.27 ± 2.42 ^a^	18.75 ± 0.25 ^ab^	43.97 ± 4.24 ^b^

CK: rice growing under Cd–Pb stress without melatonin treatment; S50: rice growing under Cd–Pb stress and sprayed with 50 μmol L^−1^ exogenous melatonin; S100: rice growing under Cd–Pb stress and sprayed with 100 μmol L^−1^ exogenous melatonin; S200: rice growing under Cd–Pb stress and sprayed with 200 μmol L^−1^ exogenous melatonin; S400: rice growing under Cd–Pb stress and sprayed with 400 μmol L^−1^ exogenous melatonin; R100: rice growing under Cd–Pb stress and irrigated with 100 μmol L^−1^ exogenous melatonin; R300: rice growing under Cd–Pb stress and irrigated with 300 μmol L^−1^ exogenous melatonin; R500: rice growing under Cd–Pb stress and irrigated with 500 μmol L^−1^ exogenous melatonin. Means sharing same letters indicate no significant difference at *p* < 0.05 according to the LSD test.

**Table 5 antioxidants-11-00776-t005:** Effects of exogenous melatonin application on soil chemical properties.

Treatment	SOM (g kg^−1^)	pH	N (g kg^−1^)	P (g kg^−1^)	K (g kg^−1^)	Cd (mg kg^−1^)	Pb (mg kg^−1^)
XCK	8.32 ± 0.08	6.42 ± 0.02	0.77 ± 0.11	1.52 ± 0.11	6.47 ± 0.29	1.15 ± 0.01	4.45 ± 0.06
XR300	8.64 ± 0.13	6.25 ± 0.03	0.83 ± 0.01	0.95 ± 0.03	6.34 ± 0.11	1.18 ± 0.02	4.59 ± 0.08
XS50	9.58 ± 0.16	6.18 ± 0.02	1.39 ± 0.05	1.64 ± 0.09	6.12 ± 0.28	1.18 ± 0.01	4.57 ± 0.04
YCK	9.82 ± 0.25	5.84 ± 0.03	1.12 ± 0.04	0.97 ± 0.12	6.81 ± 0.68	1.20 ± 0.01	3.51 ± 0.14
YR300	8.97 ± 0.09	6.86 ± 0.02	0.68 ± 0.07	0.81 ± 0.02	5.57 ± 0.20	1.21 ± 0.02	4.49 ± 0.04
YS50	9.96 ± 0.12	6.25 ± 0.03	1.47 ± 0.04	0.86 ± 0.04	6.35 ± 0.18	1.19 ± 0.01	3.97 ± 0.05

XCK: Xiangyaxiangzhan growing under Cd–Pb stress without melatonin treatment; XR300: Xiangyaxiangzhan growing under Cd–Pb stress and irrigated with 300 μmol L^−1^ exogenous melatonin; XS50: Xiangyaxiangzhan growing under Cd–Pb stress and sprayed with 50 μmol L^−1^ exogenous melatonin; YCK: Yuxiangyouzhan growing under Cd–Pb stress without melatonin treatment; YR300: Yuxiangyouzhan growing under Cd–Pb stress and irrigated with 300 μmol L^−1^ exogenous melatonin; YS50: Yuxiangyouzhan growing under Cd–Pb stress and sprayed with 50 μmol L^−1^ exogenous melatonin.

**Table 6 antioxidants-11-00776-t006:** Effects of exogenous melatonin application on bacterial diversity indexes.

Sample ID	OTU	ACE	Chao1	Simpson	Shannon	Coverage
XCK	1827.33 ± 3.93 ^ab^	1840.14 ± 4.16 ^a^	1842.1881 ± 5.4287 ^a^	0.0026 ± 0.0001 ^b^	6.6893 ± 0.0210 ^a^	0.9987 ± 0.0001 ^a^
XR300	1822.67 ± 1.20 ^b^	1837.18 ± 2.64 ^a^	1839.7701 ± 4.3173 ^a^	0.0029 ± 0.0000 ^a^	6.6352 ± 0.0065 ^b^	0.9986 ± 0.0002 ^a^
XS50	1836.33 ± 3.71 ^a^	1847.25 ± 3.07 ^a^	1851.9076 ± 4.0648 ^a^	0.0026 ± 0.0001 ^b^	6.7050 ± 0.0136 ^a^	0.9988 ± 0.0001 ^a^
YCK	1810.67 ± 7.26 ^b^	1830.41 ± 5.17 ^b^	1839.0321 ± 5.2287 ^a^	0.0030 ± 0.0001 ^a^	6.5805 ± 0.0227 ^b^	0.9984 ± 0.0002 ^a^
YR300	1834.67 ± 3.18 ^a^	1847.36 ± 4.37 ^a^	1851.8427 ± 4.0325 ^a^	0.0026 ± 0.0001 ^b^	6.7054 ± 0.0187 ^a^	0.9987 ± 0.0001 ^a^
YS50	1825.00 ± 4.16 ^ab^	1838.62 ± 2.97 ^ab^	1843.3003 ± 2.0112 ^a^	0.0027 ± 0.0001 ^b^	6.6682 ± 0.0056 ^a^	0.9986 ± 0.0002 ^a^

XCK: Xiangyaxiangzhan growing under Cd–Pb stress without melatonin treatment; XR300: Xiangyaxiangzhan growing under Cd–Pb stress and irrigated with 300 μmol L^−1^ exogenous melatonin; XS50: Xiangyaxiangzhan growing under Cd–Pb stress and sprayed with 50 μmol L^−1^ exogenous melatonin; YCK: Yuxiangyouzhan growing under Cd–Pb stress without melatonin treatment; YR300: Yuxiangyouzhan growing under Cd–Pb stress and irrigated with 300 μmol L^−1^ exogenous melatonin; YS50: Yuxiangyouzhan growing under Cd–Pb stress and sprayed with 50 μmol L^−1^ exogenous melatonin. Means sharing similar letters indicate no significant difference at *p* < 0.05 according to the LSD test.

**Table 7 antioxidants-11-00776-t007:** Effects of exogenous melatonin application on the top five dominant bacteria phyla.

Treatment	Acidobacteria	Actinobacteria	Chloroflexi	Gemmatimonadetes	Proteobacteria
XCK	0.150 ± 0.004 ^a^	0.069 ± 0.001 ^b^	0.317 ± 0.014 ^a^	0.034 ± 0.001 ^a^	0.320 ± 0.011 ^a^
XR300	0.149 ± 0.005 ^a^	0.082 ± 0.002 ^a^	0.344 ± 0.002 ^a^	0.035 ± 0.001 ^a^	0.276 ± 0.004 ^b^
XS50	0.153 ± 0.002 ^a^	0.065 ± 0.004 ^b^	0.318 ± 0.010 ^a^	0.034 ± 0.003 ^a^	0.316 ± 0.004 ^a^
YCK	0.163 ± 0.005 ^a^	0.059 ± 0.007 ^a^	0.357 ± 0.009 ^a^	0.033 ± 0.002 ^a^	0.290 ± 0.011 ^b^
YR300	0.134 ± 0.004 ^b^	0.063 ± 0.005 ^a^	0.323 ± 0.015 ^a^	0.028 ± 0.001 ^a^	0.325 ± 0.009 ^a^
YS50	0.165 ± 0.005 ^a^	0.059 ± 0.005 ^a^	0.339 ± 0.005 ^a^	0.032 ± 0.001 ^a^	0.296 ± 0.002 ^ab^

XCK: Xiangyaxiangzhan growing under Cd–Pb stress without melatonin treatment; XR300: Xiangyaxiangzhan growing under Cd–Pb stress and irrigated with 300 μmol L^−1^ exogenous melatonin; XS50: Xiangyaxiangzhan growing under Cd–Pb stress and sprayed with 50 μmol L^−1^ exogenous melatonin; YCK: Yuxiangyouzhan growing under Cd–Pb stress and without melatonin; YR300: Yuxiangyouzhan growing under Cd–Pb stress and irrigated with 300 μmol L^−1^ exogenous melatonin; YS50: Yuxiangyouzhan growing under Cd–Pb stress and sprayed with 50 μmol L^−1^ exogenous melatonin. Means sharing similar letters indicate no significant difference at *p* < 0.05 according to the LSD test.

## Data Availability

Data is contained within the article and Appendix A.

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
