# Peer review of "Effect of Exogenous Melatonin Application on the Grain Yield and Antioxidant Capacity in Aromatic Rice under Combined Lead–Cadmium Stress"

_antioxidants, 2022, doi:10.3390/antiox11040776_

Round 1
Reviewer 1 Report
I reviewed article „Exogenous melatonin application regulate the antioxidant capacity, metabolites, metal contents, grain yield, and grain quall it in fragrant rice under combined lead-cadmium stress “submitted to the journal Antioxidant.
Firstly, I think that Title is inappropriate. It suggests that melatonin is the main factor which regulate antioxidant capacity, metabolites, metal contents, grain yield, and grain quality what is not correct. Also title is too long. This should be shortened and changed.
I think that fragrant rice is not official common name of the variety, authors should check this. Is this the same as aromatic rice?
References are not in accordance with Antioxidant Instruction for authors.
L 219. „2.3.6 Determination of fragrance... “should be changed into „2.3.6 Determination of volatile compounds... “To be honest, I do not know what will mean „physiological attributes “of fragrance? In this section authors put method for Proline and pyrroline-5-carboxylic acid (P5C) contents and I do not understand this why here.
L 251. 2.3.8 Metabolite profiling. What that mean? Which metabolites? How they prepare samples? What they determinate? How they did data analysis,
Authors need to reorganise MM section into into meaningful wholes and add all important information. This section is very poorly prepared.
Figures and Tables are in a separate section. They should be inserted in a text next to where they are first time mention. Now is confusing and hard to follow. Authors use a lots of diagrams obtained from statistical software what make the whole section hard to follow and useless. Results should be clearly presented and statistical analysis used on proper way.
Overall, topic of the manuscript is not novel (melatonin is well known compounds which influence response of plants to Cd stress) but it seems to me that experiment is properly conducted. But manuscript suffer from serious drawback, especially in results and material and method sections which had to be significantly improved and manuscript should be reevaluated .
Reviewer 2 Report
he work is of interest, especially from a translational point of view, and the paper is well written and easy to read.
The methodology employed is ample and diverse. Lot of work and data, congratulations. Probably the authors should justify why the spraying and irrigation treatments as well as the MT doses employed.
I have got doubts about the novelty. As well introduced in the paper, the topic has been investigated in other plant species or rice varieties, so only the use of fragrant rice varieties is novel. Is there any difference between the fragrant rice varieties and other type of rice or plant spp?
- 79 Melatonin, known as N-acetyl-5-methoxytryptamine, is an indole tryptamine … For compounds there are common and chemical nomenclatures, so it should be the opposite. The chemical name, known as common name.
In M&M it is not clear if the experiment was carried out under natural or growth chamber conditions. Indicate photoperiod, illumination, temperature, humidity.
The huge amount of data generated are correctly presented. Have you considered the multivariant (PCA or PLS) statistical analysis of the whole sets of data, in order to see which variables are more affected?
The differences between MT doses and method of application should be deeper discussed. By the way, did you observe morphometric changes at high MT doses? In definitive, how the doses more than the compound determine the plant response? You only focus those data that show the positive effect of MT. There must be more discussion and less description, at least in the discussion section. This section seems somehow a review of the topic.
Round 2
Reviewer 1 Report
The manuscript is significanlty improved in comparison with first round. Still I have some comments:
You should add detail about how you perform data analysis in Determination of primary metabolites section. Did you have a standards? If not you just annotated compounds, not identified. If the compounds are just annotated the whole section about up and down regulation of the biosynthetic pathways are questenable. In a MM section you stated "Metabolite quantification is accomplished by multiple reaction monitoring (MRM) 262
analysis using triple quadrupole mass spectrometry" what will mean that you just annotated compounds using MS spectars? This may be incorrectly.
Also for high quality KEGG analysis you should have samples identified using standards and accompanied with transcriptomic data. Otherwise this are all just speculations. So my recommendation is to remove 3.6.3 Pathway analysis section from the manuscript, or you validate data using transcriptomic.
Author Response
The manuscript is significanlty improved in comparison with first round. Still I have some comments:
-Response: Thank you for reviewing again.
You should add detail about how you perform data analysis in Determination of primary metabolites section. Did you have a standards? If not you just annotated compounds, not identified. If the compounds are just annotated the whole section about up and down regulation of the biosynthetic pathways are questenable. In a MM section you stated "Metabolite quantification is accomplished by multiple reaction monitoring (MRM) 262 analysis using triple quadrupole mass spectrometry" what will mean that you just annotated compounds using MS spectars? This may be incorrectly.
-Response: Thanks for your reminder. We have added how to perform data analysis in the section on the determination of primary metabolites. We have revised this section for deficiencies.
Multiple reaction monitoring (MRM) scanning mode is used to identify metabolites in the samples in high-efficiency batch, so as to obtain more complete and accurate metabolic spectrum information. In the MRM mode, the quadrupole first filters the precursor ions (precursor ions) of the target substance, and excludes ions corresponding to other molecular weight substances to initially eliminate interference; the precursor ions are fragmented after the induced ionization of the collision chamber to form many fragment ions, fragment ions Then select a characteristic fragment ion by triple quadrupole filtering to eliminate the interference of non-target ions, so that the quantification is more accurate and the repeatability is better. After obtaining the metabolite mass spectrometry data of different samples, the peak area of all substance mass peaks was integrated, and the peaks of the same metabolite in different samples were integrated and corrected.
Thank you again for pointing out the error. We have improved and amended the section.
Also for high quality KEGG analysis you should have samples identified using standards and accompanied with transcriptomic data. Otherwise this are all just speculations. So my recommendation is to remove 3.6.3 Pathway analysis section from the manuscript, or you validate data using transcriptomic.
-Response: Thank you very much for your advice. We have removed the ‘3.6.3 Pathway analysis’ section from the revised manuscript.
Round 3
Reviewer 1 Report
Manuscript is improved.